# Transcriptomic Analysis of Mineralized Adipose-Derived Stem Cell Tissues for Calcific Valve Disease Modelling

**DOI:** 10.3390/ijms25042291

**Published:** 2024-02-14

**Authors:** Alyssa Brodeur, Vincent Roy, Lydia Touzel-Deschênes, Stéphanie Bianco, Arnaud Droit, Julie Fradette, Jean Ruel, François Gros-Louis

**Affiliations:** 1Department of Surgery, Faculty of Medicine, Université Laval, Quebec City, QC G1V 5C3, Canada; alyssa.brodeur.1@ulaval.ca (A.B.); vincent.roy@crchudequebec.ulaval.ca (V.R.); lydia.t-deschenes@crchudequebec.ulaval.ca (L.T.-D.); julie.fradette@fmed.ulaval.ca (J.F.); 2Division of Regenerative Medicine, CHU de Quebec Université Laval Research Centre, Quebec City, QC G1J 5B3, Canada; jean.ruel@gmc.ulaval.ca; 3Department of Molecular Medicine, CHU de Quebec Université Laval Research Centre, Quebec City, QC G1V 5C3, Canada; stephanie.bianco@crchudequebec.ulaval.ca (S.B.); arnaud.droit@fmed.ulaval.ca (A.D.); 4Computational Biology Laboratory, CHU de Quebec Université Laval Research Centre, Quebec City, QC G1V 4G2, Canada; 5Department of Mechanical Engineering, Faculty of Sciences and Engineering, Laval University, Quebec City, QC G1V 0A6, Canada

**Keywords:** tissue engineering, adipose-derived stromal/stem cells, RNA sequencing, calcific aortic valve disease, disease modelling, extracellular matrix

## Abstract

Calcific aortic valve disease (CAVD) is characterized by the fibrosis and mineralization of the aortic valve, which leads to aortic stenosis and heart failure. At the cellular level, this is due to the osteoblastic-like differentiation of valve interstitial cells (VICs), resulting in the calcification of the tissue. Unfortunately, human VICs are not readily available to study CAVD pathogenesis and the implicated mechanisms in vitro; however, adipose-derived stromal/stem cells (ASCs), carrying the patient’s specific genomic features, have emerged as a promising cell source to model cardiovascular diseases due to their multipotent nature, availability, and patient-specific characteristics. In this study, we describe a comprehensive transcriptomic analysis of tissue-engineered, scaffold-free, ASC-embedded mineralized tissue sheets using bulk RNA sequencing. Bioinformatic and gene set enrichment analyses revealed the up-regulation of genes associated with the organization of the extracellular matrix (ECM), suggesting that the ECM could play a vital role in the enhanced mineralization observed in these tissue-engineered ASC-embedded sheets. Upon comparison with publicly available gene expression datasets from CAVD patients, striking similarities emerged regarding cardiovascular diseases and ECM functions, suggesting a potential link between ECM gene expression and CAVDs pathogenesis. A matrisome-related sub-analysis revealed the ECM microenvironment promotes the transcriptional activation of the master gene runt-related transcription factor 2 (*RUNX2*), which is essential in CAVD development. Tissue-engineered ASC-embedded sheets with enhanced mineralization could be a valuable tool for research and a promising avenue for the identification of more effective aortic valve replacement therapies.

## 1. Introduction

Calcific aortic valve disease (CAVD) is a common cardiovascular condition worldwide, characterized by the progressive mineralization and fibrosis of the aortic valve, leading to stenosis and heart failure [1]. CAVD is a complex and multifaceted disease with a pathophysiology that involves a series of progressive changes in the aortic valve and surrounding tissues. Valve interstitial cells (VICs) play a significant role in the pathophysiology of CAVD and undergo osteogenic differentiation [2]. These cells also contribute to the fibrotic changes in the valve leaflets, leading to the thickening and stiffening of the tissue. VICs are also involved in the regulation of the extracellular matrix (ECM) content in the valve [3]. In CAVD, there is an imbalance in ECM turnover, with the excessive deposition of collagens, proteoglycans, and glycosaminoglycans alongside the increased activity of matrix metalloproteinases [4]. This disease represents a significant healthcare burden, particularly in the aging population. While CAVD is a well-recognized condition, understanding its underlying pathology and finding effective treatments remain substantial challenges. At the moment, there are no effective pharmacotherapies to treat this disorder, and surgical valve replacement is the only option for patients [5].

One of the primary research problems in advancing knowledge on CAVD is the limited availability of diseased valve tissues and affected cells, such as VICs [6]. To address this issue, researchers are turning to patient-derived tridimensional (3D) models and tissue engineering approaches [7]. Although current tissue-engineered heart valves (TEHVs) are primarily developed as alternatives for valve replacement, they also hold promise in disease modelling to provide a deeper understanding of the underlying CAVD pathology and to shed light on the mechanisms and potential therapies for this disease [8]. While the production of patient-specific VICs differentiated from induced pluripotent stem cells (iPSCs) may represent a good alternative to study CAVD, this approach is expensive, tedious, and time-consuming, which may limit the use of iPSCs to model CAVD [9]. Interestingly, adipose-derived stromal/stem cells (ASCs), readily accessible through a small liposuction procedure or excised subcutaneous adipose tissue biopsy, have proven their relevance for tissue-engineering endeavours and have emerged as a promising cell source due to their multipotent nature, availability, and patient-specific characteristics [10]. Furthermore, these mesenchymal cells are known to secrete various bioactive molecules that act on the critical pathways stimulating tissue regeneration [11]. 

While ASCs cultured in vitro exhibit limited valve-like properties, their use in producing tissue-engineered models offers several advantages to study CAVD. Indeed, it has been shown that ASCs can be differentiated towards osteogenic lineages and are capable of efficiently secreting an abundant collagen- and osteocalcin-rich ECM upon long-term stimulation with ascorbic acid [12,13]. Interestingly, it has been proposed that ASCs exhibit similar ECM biomarker expression to human VICs and would be a suitable cell source for TEHVs [14]. Furthermore, it has been shown that ASCs’ osteogenic differentiation closely mimics VICs osteogenic expression markers, such as matrix mineralization proteins as well as different signalling pathways [8,15,16,17].

Building upon this, we propose that the mineralized tissues produced from ASCs will serve as a suitable model for calcific valve disease. In this study, we describe a comprehensive transcriptomic analysis of scaffold-free mineralized ASC-embedded ECM sheets (ASC-sheets), which predict in silico functions linked to ECM mineralization and organization. Upon comparison with publicly available gene expression datasets from CAVD patients, similarities with ASC-sheets emerge in terms of the modulation of ECM organization and structure. From the analysis of the dysregulated matrisome components, which comprises an extensive list of ECM-related genes, a discernible microenvironment predicting the modulation of runt-related transcription factor 2 (RUNX2) targets has surfaced. Tissue-engineered ASC-sheets with enhanced mineralization could be a valuable tool for research to better understand CAVD for the development of more effective pharmacological therapies. 

## 2. Results

### 2.1. Mineralization of 3D ASC Tissue-Engineered ECM Sheets

Mineralization was induced in ASC-sheets by adding soluble factors to the differentiation medium as illustrated (Figure 1A). To visualize the protocol’s efficiency, mineralization was first qualitatively observed following alizarin red staining (ARS), a commonly used stain to identify calcium-containing mesenchymal cells in differentiated culture [18] (Figure 1B). On average, the induced and mineralized sheets displayed 42 times (*p* = 0.0079) more calcification than the corresponding uninduced sheets following ARS and DNA quantification (Figure 1C). Note that the induction of mineralization efficacy was population-dependent and varied between donors (Figure 1B,C), with the ASC4 population exhibiting lower mineralization and the ASC5 population exhibiting higher mineralization. A single ASC-sheet can be observed by Masson’s trichrome staining, revealing ASCs embedded in a 3D self-secreted ECM, as illustrated in the Appendix A.

### 2.2. Transcriptomic Analysis of 3D ASC Tissue-Engineered ECM Sheets

The mineralized and uninduced tissue-engineered ASC-sheets were then subjected to transcriptomic analysis using bulk RNA-seq. Principal component analysis (PCA) revealed distinct global gene expression clusters depending on the tested conditions (Figure 2A). Interestingly, the global gene expression measured for the uninduced condition was more tightly clustered, which is indicative of a greater homogeneity among gene transcripts. The gene expression measured in the mineralized ASC-sheets is more heterogenous, indicating that the mineralization efficacy may be population-dependent and be influenced by other factors such as the patient’s genomic background [19,20]. A total of 1059 differentially expressed genes (DEGs), with an absolute fold change of 1.5 or greater and an adjusted *p*-value of less than 0.05, were found to be significantly modulated in mineralized sheets (Appendix A), of which 583 and 476 were significantly up-regulated and down-regulated, respectively (Figure 2B). A heatmap was generated to illustrate the DEGs in the sheets produced from five different ASC populations in comparison to uninduced sheets from the same ASC populations (Figure 2C). Correlating with the ARS results, the ASC4 mineralized sheets interestingly displayed a less pronounced gene modulation profile compared to other ASC populations. 

To help interpret the gene expression data, gene set enrichment analyses (GSEA), based on the functional annotation of the DEGs, were performed. Biological processes from the gene ontology (GO) knowledgebase for up-regulated genes predicted to be enriched were strongly linked to ribosome biogenesis (GO:0042254, adjusted *p* = 1.02 × 10^−34^) as well as to ECM organization (GO:0030198, adjusted *p* = 1.83 × 10^−5^) (Figure 2D). A network analysis with all DEGs was also performed in combination with pathway analysis to better highlight the interaction of key components of this pathway via an in silico investigation using IPA software. Interestingly, the organization and mineralization of the ECM were predicted to be significantly activated, while the formation of ribosomes, a biogenesis process known to be influenced by osteoblast differentiation [21], was predicted to be inhibited (Figure 2E). Among the most prominent interacting genes, *BMP2* seems the most interconnected gene associated with ECM mineralization. 

### 2.3. Comparative Transcriptomic Analysis with CAVD Patients Dataset

We then performed a transcriptomic comparison analysis between our dataset and other publicly available transcriptomic data generated from patients’ calcified valve tissues that were previously published by Greene and collaborators [22]. While only comparing DEGs (absolute fold change > 1.5; adjusted *p* < 0.05), 284 genes were found to be commonly shared between both datasets (Figure 3A). GSEAs of these common genes, using GO biological process predictions, were more significantly linked to ECM organization (GO:0030198; adjusted *p* = 2.06 × 10^−7^, GO:0043062; adjusted *p* = 2.06 × 10^−7^) and bone development (GO:0001503, adjusted *p* = 0.002) (Figure 3B). Interestingly, when using common DEGs, GSEAs predicted cardiac diseases (arrhythmogenic right ventricular cardiomyopathy (map05412, adjusted *p* = 2.68 × 10^−4^) from the KEGG pathway and arrhythmogenic right ventricular cardiomyopathy (WP3195, adjusted *p* = 3.75 × 10^−5^) from the WikiPathway) and ECM organization (R-HSA-1474244, adjusted *p* = 3.28 × 10^−6^, from the Reactome) amongst the more significantly enriched pathways (Figure 3C). We subsequently performed an in-depth analysis of functionally enriched pathways using IPA and transcriptomic RNA-seq data generated from mineralized ASC-sheets (Appendix A). Interestingly, the DEGs were found to be linked to different valvulopathies including the calcification of the aortic valve.

### 2.4. ECM Modulation and Comparison to the Human Matrisome Database

To provide a deeper understanding of the ECM biological functions associated with the mineralization of the ASC-sheets, a sub-analysis was performed with common genes from our dataset and the human matrisome database, which is an open and comprehensive ECM atlas of all ECM-encoded genes [23]. A majority of the genes in all matrisome categories were found to be quantified by RNA-seq (Figure 4A), indicating the overall importance of the matrisome in the mineralization process. A heatmap was used to visualize the clustering of significantly modulated genes for each matrisome category (Figure 4B). We performed a functional enrichment pathway analysis using the differentially up-regulated matrisome genes (Figure 4C). Interestingly, the Hippo pathway (map04390, *p* = 6.35 × 10^−4^) and TGF-β signalling (map04350, *p* = 0.004), two major pathways involved in osteoblastic differentiation, were predicted to be strongly activated by KEGG pathway. Osteoblast differentiation (WP4787, *p* = 0.007) and BMP signalling (WP211, *P* = 0.01) were also significantly enriched according to WikiPathway. Furthermore, BMP signalling (R-HSA-201451, *p* = 3.63 × 10^−4^) and TGF-β signalling (R-HSA-170834, *p* = 0.007) were highlighted by the Reactome pathway knowledgebase. Interestingly, three distinct pathways implicating RUNX2, a specific transcription factor implicated in the differentiation and maturation of osteoblasts and of crucial importance in bone formation and growth, were detected (Transcriptional regulation by RUNX2, R-HSA-8878166, *p* = 2.53 × 10^−4^; RUNX2 regulates bone development, R-HSA-8941326, *p* = 4.79 × 10^−4^, and RUNX2 regulates chondrocyte maturation, R-HSA-8941284, *p* = 0.005).

### 2.5. Transcriptional Activation by the Master Gene RUNX2 in Mineralized Sheets

Following the GSEA using the DEGs related to the matrisome, we further specifically investigated the regulation of transcription factors in our dataset using TRRUST, an expended reference database of human transcriptional regulatory interactions [24]. We identified several transcription factors predicted to regulate DEGs, with the top 20 closely linked to osteoblastic differentiation, including RUNX2 (*p* = 0.001) (Figure 5A). 

We then constructed the transcriptional factor regulatory network of RUNX2 based on the JASPAR database (http://jaspar.genereg.net/, accessed on 14 July 2023), a curated open access database containing non-redundant transcription factor binding profiles [25]. From the 1448 predicted targets of RUNX2 in this database, 1293 were properly quantified in our dataset. The heatmap of RUNX2 targets shows the 40 significantly up-regulated and 33 down-regulated genes in the mineralized ASC-sheets compared to the uninduced sheets (Figure 5B). We also assessed the relative gene expression of *RUNX2* by RT-qPCR in ASC-sheets over time following osteoblastic differentiation induction (Figure 5C). While the expression of *RUNX2* appears to be relatively low during the initial differentiation phase, its expression was found to be stable from day 10 to 28. We finally constructed a regulatory interactome of functionally enriched DEGs involved in the activation of RUNX2 according to the IPA prediction based on RNA-seq data (Figure 5D). Interestingly, *BMP2*, which was previously identified to be a key gene associated with ECM mineralization (Figure 2E), was also found to be an important upstream regulator of RUNX2 transcriptional activation.

## 3. Discussion

We demonstrated that 3D mineralized tissue-engineered ASC-embedded ECM sheets serve as an excellent model for studying CAVD in vitro, primarily attributed to the ECM’s favourable microenvironment that facilitates the transcriptional activation of the master gene *RUNX2*. The osteoblastic differentiation of ASC resulted in the generation of manipulatable mineralized ECM sheets exhibiting a distinctive transcriptomic signature compared to uninduced sheets. A bioinformatic analysis and GSEA revealed the implication of the mineralization and modulation of the ECM as an important biological process. Upon comparison with other publicly available gene sets, striking similarities emerged between patients’ calcified valve tissues. Indeed, modulated genes found in mineralized ASC-sheets were comparable to those found in an RNA-seq dataset generated from patients’ mineralized valves [22]. The pathway analysis strongly connected common DEGs for both datasets to cardiomyopathies [26]. Of particular interest, hypertrophic cardiomyopathy is the most prevalent primary cardiomyopathy of a genetic cause and is associated with more severe CAVD and coronary artery disease [27]. It is also a common comorbidity and consequence of aortic stenosis, hinting to a link between both pathologies [28]. On the other hand, the GSEA of GO biological processes predicted the involvement of the ECM and extracellular structure organization, suggesting a possible biological relationship between such cardiovascular diseases that is related to ECM organization. In osteoblastic differentiation, the ECM not only provides structure for mineralization, it also helps with commitment toward the osteoblast lineage, as well as the storage and release of growth factors, such as BMP2 [21,29], a known activator of osteoblast differentiation through RUNX2-dependent expression [30]. Indeed, the BMP signalling pathway and RUNX2 are known to cooperatively interact during osteogenesis [31].

The GSEA and in silico pathway analysis of mineralized ASC-sheets predicted the significant inhibition of ribosome biogenesis, a process known to be dynamically regulated in osteoblast differentiation [32]. Interestingly, *RUNX2*, which is predicted to regulate transcription according to our results, is known to repress ribosome formation by binding to the promoter for ribosomal RNA (rRNA) genes during the early phase of osteoblast differentiation [33,34]. Our results indicate that the ECM microenvironment produced by ASCs following the induction of mineralization allowed for transcriptional activation or inhibition by *RUNX2*. In addition to its crucial role in bone-like development and osteoblastic differentiation, *RUNX2* transcripts are also found to be overexpressed in patients with CAVD [35]. In this disease, it has been shown that VICs can acquire an osteoblast-like phenotype associated with an increased RUNX2 and BMP2 expression [36,37]. Our in vitro CAVD model therefore recapitulated the regulation of osteoblast differentiation through RUNX2, which influenced the level of rRNA gene transcription, suggesting a strong mechanistic link between RUNX2, ribosome biogenesis, and osteogenesis. It also confirmed that the transcriptional activation of RUNX2 by a specific ECM microenvironment may inherently contribute to VIC mineralization and to CAVD. While our model allows for a better understanding of the key mechanistic pathways in CAVD, the latter remains a multifaceted and complex disease involving other not yet fully understood pathophysiological mechanisms such as chronic inflammation, oxidative stress, endothelial dysfunction, hemodynamic factors, as well as genetic factors that may predispose individuals to CAVD [5]. Research in this field is ongoing, and advancements in molecular biology and genetics continue to provide new insights into the pathophysiology of this condition. The generation of such patient-derived ASC-embedded ECM sheet models in vitro have the potential to significantly enhance our understanding of CAVD. Furthermore, the use of the CRISPR-Cas9 gene-editing tool in such a personalized model could also allow researchers to investigate the functional consequences of genetic variants associated with CAVD, helping to unravel the underlying molecular mechanisms. 

The *RUNX2* gene plays a crucial role in osteoblast differentiation and bone formation. In the context of heart valves and CAVD, the incorporation of cells for the generation of RUNX2 knockout tissue-engineered heart valves in previously described model [38], could be a potential strategy to reduce tissue mineralization in heart valves. While these approaches offer exciting possibilities, it is important to note that challenges and ethical considerations, such as off-target effects and the unintended consequences of genetic modifications, need to be carefully addressed; however, as technology and methodologies continue to advance, the combination of personalized in vitro models and CRISPR-Cas9 holds great potential for transforming our understanding of CAVD and improving drug discovery efforts for this complex disease. Research in this area is ongoing, and scientists are actively exploring various genetic and molecular strategies to understand and potentially mitigate valve calcification in the context of CAVD.

In conclusion, we characterized an all-human tissue-engineered 3D model based on ASCs, in which the disease transcriptomic signatures can be recapitulated to study CAVD. We also showed that the ECM microenvironment established upon the induction of mineralization in ASCs led to transcriptional regulation by master gene *RUNX2*. The pathophysiology of CAVD is still an active area of research, and ongoing studies aim to better understand the molecular and cellular mechanisms involved in the disease. This model, which can be made from the patient’s own cells, offers several advantages in studying the pathophysiology of CAVD and could pave the way for tailoring therapies based on an individual’s unique genetic and molecular profile.

## 4. Materials and Methods

### 4.1. ASCs Culture

The use of human cells was approved by the ethical research board of the CHU de Quebec-Université Laval (protocol numbers: R-002-1343 and DR-002-1117), and individuals were recruited following informed consent and on a voluntary basis. Human ASCs were isolated as previously described [10]. Briefly, cells were extracted from adipose tissues of five healthy female individuals undergoing lipoaspiration procedures and that are aged between 35 to 44 years old with a body mass index ranging from 20.1 to 25.6 kg/m^2^. To recover the stromal vascular fraction containing ASCs, tissues were digested with 0.075% collagenase type 1A (Millipore-Sigma, Burlington, MA, USA) in Krebs–Ringer buffer (Millipore-Sigma) for 1 h at 37 °C followed by 10 min in 0.25% trypsin (Thermo Fisher Scientific, Waltham, MA, USA) and red cell lysis buffer in NH_4_Cl (Millipore-Sigma). ASCs were then cultured and expanded in 1:1 Dulbecco’s Modified Eagle’s Medium: Ham’s F12 medium (DMEM:F12; Invitrogen, Burlington, ON, Canada) containing 10% fetal bovine serum (FBS; VWR, Radnor, PA, USA), 100 IU/mL penicillin (Millipore-Sigma) and 25 μg/mL gentamicin (Gemini Bio, West Sacramento, CA, USA). Cells were cryopreserved, thawed, expanded, and cultured on Nunc^TM^ dishes (VWR), and grown in a 37 °C humidified incubator with 8% CO_2_, and the medium was changed three times a week. Upon flow cytometry analyses, the extracted ASCs using our previously published protocol expressed the cell surface markers CD44, CD73, CD90, and CD105 while being negative for CD34 and CD45 [39].

### 4.2. ASC-Embedded Tissue-Engineered ECM Sheet Production

The production of 3D mineralized tissue-engineered ASC-embedded ECM sheets was achieved upon the induction of osteoblastic differentiation as previously described [12,40]. Briefly, cells were seeded at a density of 5200 cells/cm^2^ in six-well plates with an anchoring ring-shaped paper to limit tissue contraction and kept in culture in ECM deposition culture media containing DMEM:F12 (Invitrogen), 10% FBS (VWR), an antibiotic cocktail of penicillin (Millipore-Sigma)/gentamycin (Gemini Bio), and 50 µg/mL freshly prepared L-ascorbic acid (Millipore-Sigma) to favour the self-secretion and deposition of ECM proteins as previously described [41]. For a subset of the cultures, at day 3, media was changed to osteoblastic differentiation media, which consisted of the above-described ECM deposition culture media in which 3.6 mM calcium chloride (Millipore-Sigma), 10 nM dexamethasone (Millipore-Sigma), 10 nM 1α,25-dihydroxyvitamin D3 (Milipore-Sigma), and 50 μM ascorbate-2-phosphate (Millipore-Sigma) were added. At day 10, 3.5 mM ß-glycerophosphate (Millipore-Sigma) was added to this differentiation media for the rest of the culture. Both the ECM deposition media (uninduced cultures) and osteoblastic differentiation media (mineralized cultures) were changed three times a week, and cells were kept in culture for a total of 28 days for 3D ASC-sheet production (Figure 1A).

### 4.3. Quantification of Mineralization by Alizarin Red Staining

Total sheet mineralization was quantified by ARS. After 28 days of culture, ASC-sheets were fixed with 3.7% formalin (ChapTec, Montreal, QC, Canada) for 20 min. Then, the sheets were washed with water and incubated with 40 mM alizarin sulfonate sodium (Millipore-Sigma) for 30 min under constant agitation. Sheets were washed again, air-dried, and photographed with an iPhone camera (Apple, Cupertino, CA, USA) to provide qualitative results. To accurately quantify with ARS, the sheets were incubated in 10% acetic acid (Thermo Fisher Scientific) for 30 min with agitation and then heated at 85 °C for 10 min. The soluble fraction containing the acid and the dissolved stain were neutralized with 10% ammonium hydroxide (Thermo Fisher Scientific). The pH range was confirmed to be between 4.1 and 4.5 using pH paper (Thermo Fisher Scientific) to ensure the proper quantification of the samples that were read using the microplate reader (Bio-Rad Laboratories Inc., Hercules, CA, USA) at the 405 nm wavelength. The mean concentrations were normalized per DNA content extracted from a triplicate of 6 mm punches of the mature sheets (Thermo Fisher Scientific) before being snap frozen. Genomic DNA was extracted using a DNeasy Blood and Tissue Kit (QIAGEN, Hilden, Germany), and a Quant-iT^TM^ PicoGreen assay was used as per the manufacturer’s instructions to quantify the sample double-stranded DNA (Thermo Fisher Scientific) at excitation/emission wavelengths of 485/520 nm using a Varioskan microplate reader (Thermo Fisher Scientific).

### 4.4. Bulk RNA Sequencing

A pool of three sheets for each mineralized ASC-embedded ECM sheet and uninduced sheet was homogenized in lysis buffer (10 mM Tris-HCl, 10 mM EDTA, 2% SDS, pH 9.0) containing 0.5 mg/mL Proteinase K (BioBasic Inc., Markham, ON, Canada) and incubated for 30 min at 55 °C. Genomic DNA was eliminated with QIAzol (QIAGEN) and chloroform (Thermo Fisher Scientific) before RNA precipitation with ethanol (Les Alcools de commerce, Boucherville, QC, Canada). Then, the RNeasy Mini Kit (QIAGEN) was used to isolate the total RNA following the manufacturer’s instructions. An RNA quality check (Bioanalyzer 2100, Agilent, Santa Clara, CA, USA) was performed on samples before library preparation with Illumina Stranded Total RNA Prep (Illumina, San Diego, CA, USA). The NovaSeq 6000 PE100 model with NovaSeq Reagent kits (Illumina) sequenced RNA for 25 million reads. The trimming of raw paired-end reads was performed using fastp v0.23.2 [42]. Fastqc v0.11.9 [43] and multiqc v1.12 [44] were used to ensure the quality of the reads for a quality check of the raw and trimmed data. The pseudo-alignment approach implemented in kallisto v0.48.0 [45] was used for quantification against the human reference genome (GRCh38/hg38). All R analyses used R v4.2.2 (R Foundation for Statistical Computing, Vienna, Austria, https://www.r-project.org, accessed on 18 April 2023). A differential expression analysis was performed using the DESeq2 v1.38.3 package [46]. Principal component analysis was performed using the ClustVis large data edition 1.0 online tool [47].

### 4.5. Bioinformatics and Gene Set Enrichment Analysis 

For the bioinformatic analysis of the RNA-seq, thresholds were applied with a fold change of 1.5 or greater and an adjusted *p*-value < 0.05 or *p*-value < 0.05. Gene set enrichment analyses (GSEA) were performed using various databases: the Gene Ontology (GO) Knowledgebase [48,49], Kyoto Encyclopedia of Genes and Genomes (KEGG) pathway database [50], WikiPathway [51,52] and Reactome Pathway Knowledgebase [53]. The normalized enrichment score was calculated using the online platform Web-based Gene Set Analysis Toolkit (WebGestalt) 2019 [54]. Dot plots were generated using https://www.bioinformatics.com.cn/en (accessed on 14 July 2023), a free online platform for data analysis and visualization. With the use of the online gene annotation and analysis resource Metascape [55], regulatory relationships with transcription factor regulation were obtained using the Transcriptional Regulatory Relationships Unraveled Sentence-based Text mining (TRRUST) database [56]. Datasets were then uploaded and analyzed with Ingenuity Pathway Analysis (IPA) software version 01-22-01 (QIAGEN) [57]. A comparative analysis with open access data and databases was performed on a gene set from the Bulk RNA-seq of human mineralized valve tissues and was exported from IPA software, which was previously published by Greene and collaborators [22]. Human ECM proteins and an ECM-associated protein composition and genes list were also exported from MatrisomeDB 2.0 [23]. Lastly, predicted genes targeted by RUNX2 were extracted from the Jaspar 2022 database [25], an open-access database of transcription factor binding profiles.

### 4.6. Reverse Transcription Quantitative PCR

The RNeasy Mini Kit (QIAGEN) was used to isolate the total RNA from the ASC-embedded ECM sheets at various time points of the differentiation protocol (days 0, 3, 10, 17, 21, and 28) as previously described above. Complementary DNA was produced using a QuantiTect Reverse Transcription Kit (QIAGEN). Quantitative polymerase chain reaction (PCR) was then performed for *RUNX2* (Forward: 5′-TTACTTACACCCCGCCAGTC-3′; Reverse: 5′-TATGGAGTGCTGCTGGTCTG-3′) and *GAPDH* (Forward: 5′-TGCACCACCAACTGCTTAGC-3′; Reverse: 5′-GGCATGGACTGTGGTCATGAG-3′). PowerTrack SYBR Green Master Mix (Thermo Fisher Scientific) was used following the manufacturer’s recommendations with the LightCycler 480 Real-time PCR system (Roche, Basel, Switzerland). Values were normalized using the *GAPDH* housekeeping gene.

### 4.7. Statistical Analysis

Statistical analyses were performed using the GraphPad Prism 10 software (GraphPad, San Diego, CA, USA). Mann–Whitney and two-way ANOVA with Tukey’s multiple comparisons test were performed. A *p*-value equal or lower than 0.05 was considered statistically significant. For all experiments, *N* = 5 refers to five different ASC populations, and *n* = 3 refers to the technical experimental triplicate. Refer to the figure legends for each experiment’s specific statistical detail. 

## Figures and Tables

**Figure 1 ijms-25-02291-f001:**
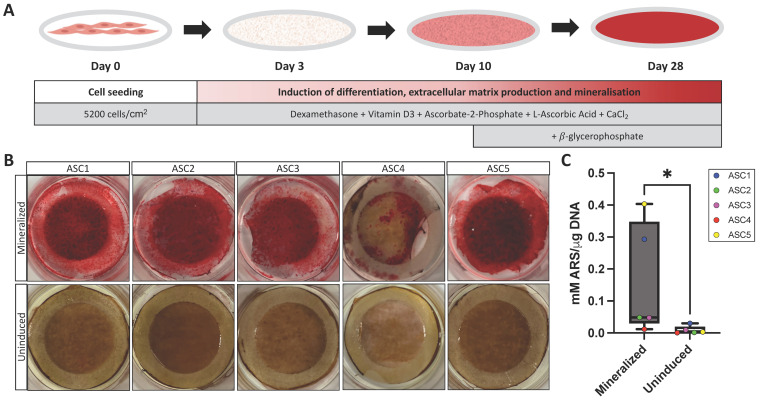
Mineralization of ASC tissue-engineered ECM sheets. (**A**) Schematic representation of the osteoblastic differentiation protocol over 28 days of culture. (**B**) Macroscopic view of mineralized and uninduced tissue-engineered sheets after ARS. (**C**) Box plot of ARS quantification (mM) with mean of the triplicate shown per cell population normalized on DNA quantity (μg). Statistics: Mann–Whitney test, *N* = 5, *n* = 3, * *p* < 0.05.

**Figure 2 ijms-25-02291-f002:**
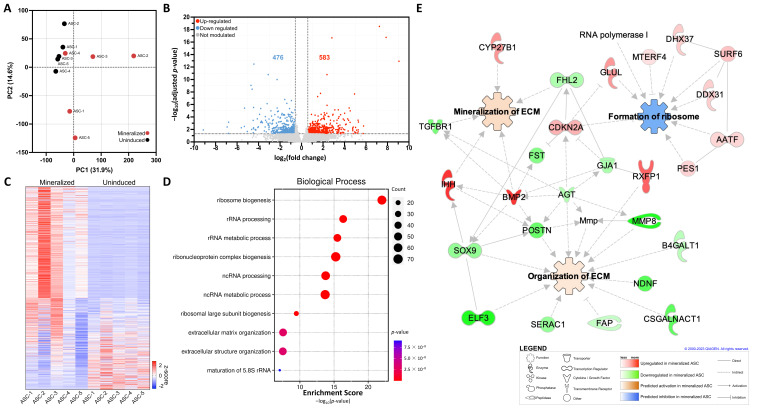
Transcriptomic overview and GSEA of mineralized ASC-embedded ECM sheets. (**A**) Cluster analysis through biplots using PCA to visualize patterns in PC1 versus PC2. (**B**) Volcano plot of DEGs from mineralized compared to uninduced ASC-sheets. (**C**) Heatmap illustrating the z-score derived from transcripts per million (TPM) values of significantly modulated genes across the five ASC populations in mineralized or uninduced sheets. (**D**) GSEA for up-regulated genes associated with biological processes in the GO database. (**E**) IPA-generated interactome of modulated genes linked with formation of ribosome, organization of ECM, and mineralization of ECM functions. *N* = 5, *n* = 3.

**Figure 3 ijms-25-02291-f003:**
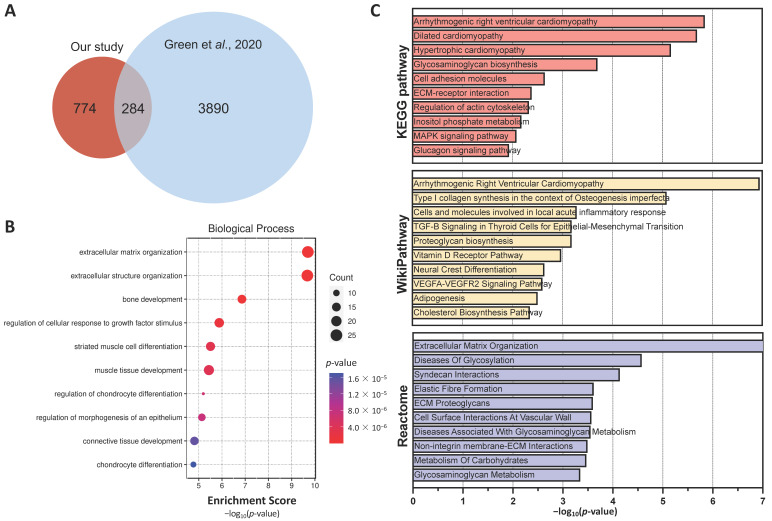
Comparative transcriptomic analysis with CAVD patient dataset. (**A**) Venn diagram representing the overlapping DEGs between our study and the study conducted by Green and collaborators [22]. (**B**) GSEA using enrichment scores from common genes between mineralized ASC-sheets and human mineralized valve tissues, focusing on GO biological processes. (**C**) GSEA with enrichment score (*p*-value) for top ten categories from KEGG pathway database, WikiPathway, and Reactome pathway knowledgebase.

**Figure 4 ijms-25-02291-f004:**
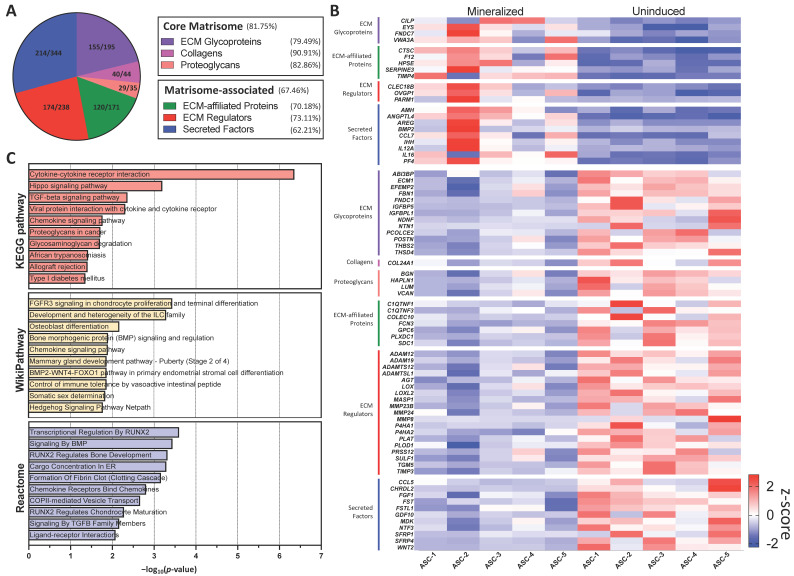
In silico prediction of ECM modulation and comparison to human matrisome. (**A**) Pie chart showing the number of core matrisome and matrisome-associated genes quantified in mineralized and uninduced sheets. (**B**) Heatmap illustrating the z-score derived from TPM values of significantly modulated matrisome-related genes across the five ASC populations in both mineralized and uninduced conditions. (**C**) GSEA of up-regulated genes with *p*-value from common genes to mineralized ASC sheets and human matrisome from KEGG pathway database, WikiPathway, and Reactome pathway knowledgebase.

**Figure 5 ijms-25-02291-f005:**
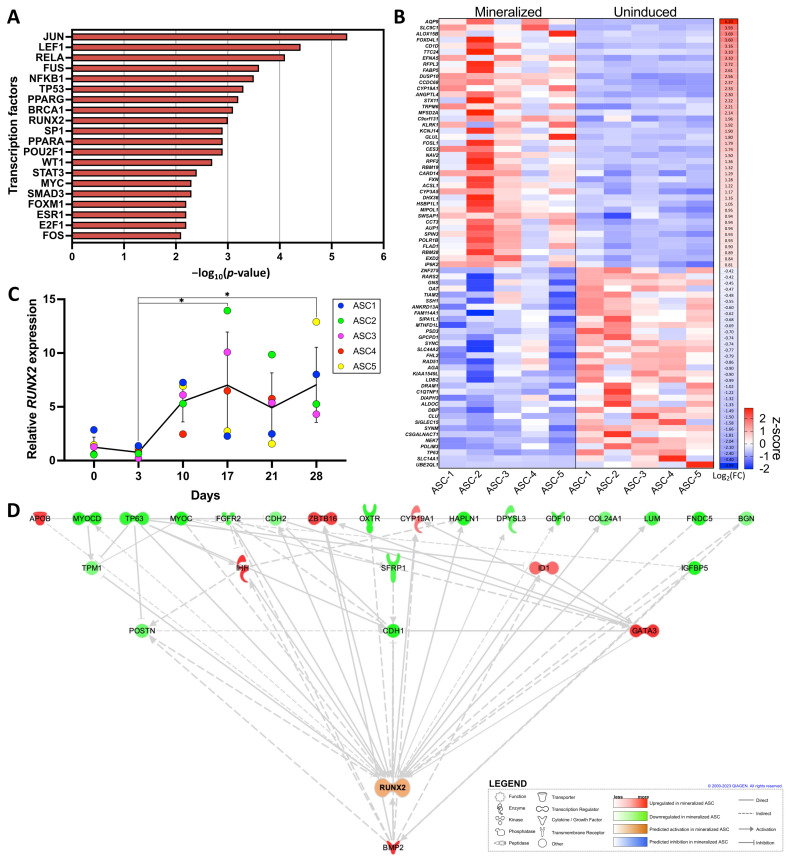
In silico prediction of transcriptomic regulation by RUNX2. (**A**) GSEA with *p*-value from up-regulated genes in mineralized ASC sheets from the TRRUST database. (**B**) Heatmap illustrating the z-score derived from TPM values of significantly modulated RUNX2 targeted genes from Jaspar database across the five ASC populations in mineralized and uninduced sheets. (**C**) Relative gene expression of *RUNX2* over time during ASC osteoblastic differentiation. *RUNX2* expression was normalized to the housekeeping gene *GAPDH.* (**D**) IPA-generated interactome of modulated genes linked to RUNX2. Statistics: two-way ANOVA with Tukey’s multiple comparisons test, *N* = 5, *n* = 3, * *p* < 0.05.

## Data Availability

Data are contained within the article and Appendix A. The raw transcriptomic data presented in this study are available on request to the corresponding author.

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
