# Peer review of "Transcriptomic Analysis of Mineralized Adipose-Derived Stem Cell Tissues for Calcific Valve Disease Modelling"

_ijms, 2024, doi:10.3390/ijms25042291_

Round 1

Reviewer 1 Report

Comments and Suggestions for Authors

In this paper, the researchers conduct a thorough transcriptomic analysis of mineralized tissue sheets embedded with adipose-derived stem cells (ASCs), which are scaffold-free.
The analysis is performed through bulk RNA sequencing. The manuscript is presented in a clear and structured manner, demonstrating relevance to the field. The experimental design aligns well with the hypothesis being tested.

Reproducibility of the manuscript's results is supported by the detailed information provided in the methods section, which involves intricate procedures. The figures, featuring graphs and images, are fitting and effectively convey the data in an easily understandable manner. The conclusions drawn are in harmony with the presented evidence and arguments. Notably, the references cited are pertinent publications, and there is an absence of self-citations.

I would suggest to include the following:

How did you establish the identity of cells obtained from lipoaspirate samples as adipose-derived stem cells (ASC)? To confirm their stemness capabilities, it is essential to observe positivity for specific markers like CD73, CD90, CD105, while also confirming the absence of other markers (CD34-, CD45-, HLA-DR-). In experiments employing ASC as a substitute for valvular interstitial cells (VIC) in an in vitro research model of calcific aortic valve disease (CAVD), it is crucial to ensure the initial purity of the cell population.

Line 317: Is the use of "8% CO2" for culture conditions accurate, or is it a typographical error? Typically, cells are cultivated in cell culture incubators with 5% CO2. If the usage of 8% CO2 is intentional, could you please provide an explanation or a reference supporting this deviation from the standard practice?

Line 319 and other places: Please substitute the term "tissue-engineered" as it does not accurately describe the structure. Instead, refer to it as a sheet of mineralized cells with extracellular matrix (ECM) deposition. The term "sheets" does not accurately represent tissue, and there hasn't been a demonstration that valvular interstitial cells (VIC) exclusively produce monolayer osteoblastic differentiation in calcific aortic valve disease (CAVD).

In my view, the study's most significant finding lies in the examination of dysregulated matrisome components, uncovering a distinct microenvironment that foreshadows the modulation of RUNX2 targets, influencing the rRNA level of gene transcription and demonstrating the link between RUNX2, ribosomes biogenesis and osteogenesis.
The augmented mineralization observed in ASC-sheets could serve as a valuable research tool, offering insights that enhance the understanding of calcific aortic valve disease (CAVD) and contribute to the development of more efficacious pharmacological therapies.

Reviewer 2 Report

Comments and Suggestions for Authors

Summary

This study investigated the transcriptome of mineralized human adipose-derived stromal/stem cells (ASCs) as a model of calcific aortic valve disease (CAVD) due to osteogenic differentiation of valve interstitial cells (VICs). Bioinformatic analysis of RNA sequencing data from ASCs in this study and comparison with data from the CAVID patient dataset and the human matrisome database revealed upregulation of extracellular matrix (ECM) genes and an osteogenic transcription factor, RUNX2, during ASC mineralization. These analyses were well performed and nicely presented, but some careless mistakes remained. Revising the minor points below will make this manuscript perfect.

This study has successfully identified the BMP2-RUNX2-ECM axis as a candidate mechanism of CAVD development due to VIC osteogenesis. It will contribute to further research for pathophysiological study and drug development. I believe this manuscript is worthy of publication in the International Journal of Molecular Sciences.

Comments

1.          Figure 2A: The contribution ratio of PC1 and PC2 must be provided.

2.          Figures 2E and 5D: The rectangles in the LEGEND (Upregulated in mineralized ASC, etc.) is not colored.

3.          Figure 3A: The Venn diagram was not shown. Revise the Figure.

4.          Figures 4B and 5B: Z-score is not colored.

5.          Figure 5B: Z-score

6.          Figure 5C: Probably “*p < 0.05” was not shown (day 3 vs day 17 and day 3 vs day 28). Check the Figure.

Minor points

7.          Line 20: “ASC” should be “ASCs”.
